# Epigenetic Modifiers: Exploring the Roles of Histone Methyltransferases and Demethylases in Cancer and Neurodegeneration

**DOI:** 10.3390/biology13121008

**Published:** 2024-12-03

**Authors:** Lauren Reed, Janak Abraham, Shay Patel, Shilpa S. Dhar

**Affiliations:** Department of Gastrointestinal Medical Oncology, The University of Texas MD Anderson Cancer Center, 1515 Holcombe Blvd., Houston, TX 77030, USA; laurenreed559@gmail.com (L.R.); janakabraham4@gmail.com (J.A.);

**Keywords:** histone modifiers, Alzheimer’s, Parkinson’s, Huntington’s, glioma, glioblastoma, medulloblastoma, oncogene, tumor suppressor

## Abstract

Histone methyltransferases (HMTs) and histone demethylases (HDMs) are enzymes that modify histones, the proteins around which DNA is wrapped. These modifications play a crucial role in regulating gene expression by adding or removing methyl groups, which can activate or silence genes. In this review, we explore the roles of HMTs and HDMs in the development of cancer and neurodegenerative diseases, specifically, Alzheimer’s, Parkinson’s, and Huntington’s diseases. Regarding cancer, HMTs and HDMs can drive tumor growth and progression by altering gene expression patterns that regulate cell proliferation and survival. For example, the dysregulation of enzymes like KMT2D (an HMT) and KDM2A (an HDM) is linked to various cancers including gastric cancer by influencing the chromatin structure and gene activity. In neurodegenerative diseases, these enzymes impact the health of neurons by modifying genes involved in brain function. In Alzheimer’s, Parkinson’s, and Huntington’s diseases, changes in HMT and HDM activity can lead to the dysregulation of genes critical for neuron survival, contributing to cognitive decline and motor dysfunction. Herein, we highlight the similarities and differences in how HMTs and HDMs function in cancer and neurodegeneration. Understanding these roles may reveal new therapeutic targets that address the epigenetic underpinnings of both cancer and neurodegenerative diseases, offering hope for innovative treatments.

## 1. Introduction

Histone lysine methylation is a posttranslational modification that can activate or deactivate gene expression. Histone methyltransferases (HMTs) and histone demethylases (HDMs) are histone lysine modifiers that carry out these changes in gene expression by strengthening or weakening the interactions between a histone and DNA (Figure 1a). HMTs and HDMs have been known to both activate and deactivate gene expression, and the effect of methylation can depend on the modification location (Figure 1b). For example, methylations on histone 3 lysine 4 (H3K4), H3K36, and H3K79 are generally considered activation marks, whereas methylations at H3K9, H3K27, and H4K20 are known to be repressive marks [1]. The methylation level can also differ, with some marks being monomethylated, dimethylated, or trimethylated by HMTs.

Additionally, HMTs and HDMs can regulate specific lysine marks that have been known to play a role in the pathogenesis of many cancers [2]. For instance, the methylation of H3K4 by KMT2D, the most frequently mutated HMT across many cancer types, weakens the interactions of histone with DNA, allowing the genes in this region to become activated [1]. When mutations of KMT2D decrease methylation at H3K4, genes that suppress the Notch pathway are repressed, leading to the development of cancers like medulloblastoma (MB) [1]. With similar findings for other cancer types, KMT2D is known as a tumor suppressor [3]. Therefore, HMTs and HDMs can be broadly classified as tumor suppressors or oncogenes. However, some histone lysine modifiers have cancer type-dependent effects.

In the present review, we focused on the roles of HMTs and HDMs identified in neuronal cancers such as glioma, glioblastoma (GBM), and MB. Gliomas are tumors that arise from glial cells in the brain and are classified into two grades and four stages. Low-grade gliomas (LGGs; stages I and II) are slow-growing tumors, whereas high-grade gliomas (HGGs; stages III and IV) appear more abnormal and proliferate at faster rates. A stage IV glioma is known as a GBM, which is the fastest growing and most aggressive glioma. GBMs are most common in patients 40 years of age or older and make up 60.2% of all gliomas. However, only 7% of patients diagnosed with GBM survive longer than 5 years, making it one of the deadliest cancers [4]. In comparison, MBs develop in the cerebellum and are the most common malignant brain tumors in children [5,6]. Current MB treatments cause severe neurological side effects [5]. Many HMTs and HDMs play crucial roles in these cancers, though some mechanisms of development remain unknown.

Additionally, many histone lysine modifiers have been associated with neurodegenerative diseases such as Alzheimer’s disease (AD), Parkinson’s disease (PD), and Huntington’s disease (HD). AD is a progressive neurodegenerative disease marked by cognitive decline and memory deficits. Although amyloid plaques and neurofibrillary tangles characterize its pathology, the primary cause of the disease is unknown, and no effective treatment exists [7]. PD is a neurological condition in which neuron degradation in the substantia nigra causes motor symptoms such as resting tremor, rigidity, and postural instability. Debilitating pain is a lesser-known symptom of PD that can greatly worsen the patient’s quality of life [8]. Whereas genetics plays a role in the pathology of PD, its exact cause is unknown. HD is a genetic neurodegenerative disease caused by an expansion in the polyglutamine region in exon 1 of the Huntingtin (*HTT*) gene located on human chromosome 4. HTT protein is ubiquitously expressed in the brain, and expansion of greater than 36 CAG repeats in the polyglutamine region creates a mutated HTT that promotes neurodegeneration in the striatum and cortex [9]. The mechanism by which mutated HTT promotes HD is unknown, and neuron breakdown can create balance and movement problems along with psychiatric disorders [10]. In these three diseases, histone modifiers are known to affect the regulation of genes involved in essential brain functions [11].

As described herein, we reviewed the roles of HMTs and HDMs in glioma, GBM, MB, AD, PD, and HD. Many of these HMTs and HDMs are found to play a role in neuronal cancers. Although HMTs and HDMs have been associated with these neurodegenerative diseases, the mechanisms explaining those associations are largely unknown. Additionally, the crossover of HMT and HDM functions with neuronal cancers and neurodegenerative diseases has remained largely unexplored. We undertook the present review to examine the roles of histone lysine modifiers in brain cancers and neurodegenerative diseases in an effort to understand the interconnections and divergences among these pathologies. We further characterize the relationship between the function of histone modifiers and brain pathologies to reveal lysine methylation as a potential target for novel brain cancer and neurodegenerative disease therapies.

## 2. Overview of HMTs and HDMs Associated with Cancer

HMTs and HDMs, which target the activator marks H3K4, H3K36, and H3K79 as well as the repressor marks H3K9, H3K27, and H4K20, were screened for their associations with cancers and neurodegenerative diseases (Appendix A). Many histone lysine modifiers affecting these marks have been implicated across multiple cancer types. Some of these HMTs and HDMs were identified as having an oncogenic, tumor suppressive, or cancer type-dependent effect on their respective marks. Specifically, modifiers like KDM2A have an oncogenic effect on many cancers, whereas others such as SETD2 and KMT2D play tumor suppressive roles.

Some histone lysine modifiers like SET-8 and SETD3 have been shown to have cancer type-dependent effects. SET-8 is an HMT that acts on H4K20, which plays a critical role in transcriptional repression and chromatin structure maintenance. In cancer cells, SET-8 is often overexpressed and has been associated with both tumor suppression and promotion, depending on the context. Its overexpression can lead to altered chromatin modifications that silence tumor suppressor genes, thereby facilitating the progression and metastasis of various cancers including breast, bladder, and prostate cancer. Conversely, in some contexts, SET-8 may act as a tumor suppressor by regulating the expression of genes involved in cell cycle control and apoptosis [12]. High levels of SETD3, an HMT that acts on H3K36, have been associated with the development of B-cell lymphoma, hepatocellular carcinoma (HCC), triple negative and p53-positive breast tumors, renal cell tumors, early-stage ovarian carcinoma, and colorectal cancer [2]. SETD3 overexpression was identified as a potential biomarker in many of these cases and correlated with poor prognosis. Although these studies characterize SETD3 as an oncoprotein, evidence demonstrates that high levels of SETD3 enhance the sensitization of cervical cancer cells to radiotherapy, potentially implicating SETD3 as a tumor suppressor [13]. Even among different cancer subtypes, the role of SETD3 differs. For example, SETD3 negatively affects relapse-free survival in patients with triple-negative and p53 breast cancer. However, in those with estrogen receptor-positive and or luminal A-type breast cancer, SETD3 has been linked with increased relapse-free survival rates [14].

Cancer type-dependent histone lysine modifiers, like SET-8 and SETD3, carry out the complex functions of HMTs and HDMs, warranting research into their roles in the diagnosis of multiple cancers and their subtypes. Although these modifiers have documented oncogenic or tumor suppressive effects on many cancers, their potential role regarding brain cancer remains largely unknown (Figure 2). For example, no studies have demonstrated the involvement of SETD3 or SET-8 in brain cancer. Therefore, research is needed to further understand the roles of these modifiers in brain cancer to develop novel therapies for it.

## 3. Roles of HMTs and HDMs in Brain Cancer

Whereas many HMTs and HDMs play a role in multiple cancer types, recent findings have shed light on the epigenetic mechanisms in brain cancer, although the function of these enzymes remains somewhat elusive. Depending on the cancer type, different brain locations can be affected (Figure 3). For example, gliomas are frequently located in the frontal and temporal lobes but have also been known to arise in the parietal lobe, occipital lobe, and deeper structures [15]. LGGs are typically located in the cerebral hemispheres in adults but are very common in the cerebellum of pediatric patients [16]. Grade IV glioma, also known as GBM, is commonly found in the cerebral lobes of the brain, especially in the frontal and temporal lobes. Although rare, GBM has also arisen in the cerebellum, spinal cord, and brainstem [17]. Whereas adult HGGs are more common in the cerebral lobes, pediatric HGGs are found throughout the central nervous system. One pediatric HGG subtype, diffuse midline glioma (DMG), develops in the midline structures of the brain, namely the thalamus, brainstem, and spinal cord [18,19]. Additionally, MB, the most common malignant brain tumor in pediatric patients, is located in the cerebellum, and different MB variants and subtypes have been identified. Many of the HMTs and HDMs explored were found to have effects on glioma, GBM, and MB. Identifying the roles of HMTs and HDMs in the pathologies above will lead to better treatment options, especially given the side effects of current brain cancer treatment.

PRDM9, a lysine methyltransferase that catalyzes methylation at the H3K4 mark, has been identified as an oncogene with broad involvement in multiple cancers including breast cancer, liver cancer, and head and neck squamous cell carcinoma [20]. Additionally, PRDM9 was shown to be involved with GBM. The enrichment of structural variants in highly recombining regions was associated with PRDM9 expression in GBM [21].

SETD1A, an HMT known for its activity on the H3K4 methylation mark, has been implicated to play a role in the development of HGGs in pediatric patients [22]. In addition to its role in gliomas, SETD1A has been found to be overexpressed in patients with breast cancer, leading to increased tumor growth. Knockout SETD1A cultures have demonstrated that its absence induces cellular senescence, effectively halting cell division and limiting tumor progression. Conversely, tumor cells expressing SETD1A exhibit enhanced proliferation, as this modifier plays a crucial role in the regulation of mitotic processes, promoting oncogenic behavior [23].

The KMT2 family of HMTs including KMT2A, KMT2C, and KMT2D catalyzes the methylation of H3K4, playing a crucial role in regulating gene expression by modifying the chromatin structure at key regulatory regions of the genome. This family of transferases has been implicated to have a role in 33 different cancers ranging from brain cancers to cancers in small extremities [24]. KMT2C and KMT2D function as tumor suppressors in many cancers such as renal cell carcinoma, acute myeloid leukemia (AML), lung adenocarcinoma, and MB [24]. Deletion or mutation of either KMT2C or KMT2D results in elevated levels of expression of KDM6A, an HDM affecting H3K27. This alteration promotes the upregulation of genes such as MMP3, which have been implicated to have a role in metastasis, especially that with a predilection for brain regions, thereby facilitating metastatic progression through enhanced KDM6A activity and subsequent changes in gene expression [3]. Moreover, the loss or mutation of KMT2D has been found to negatively impact the survival times of patients with a specific type of MB called SHH-driven MB, in which the beta subtype of the protein is affected [25]. This indicates that KMT2D offers protective effects for patients with MB and emphasizes its role as a tumor suppressor.

SET7/9, another HMT, significantly influences specific tumor-related pathways by functioning as a tumor suppressor. SET7/9 operates by mediating the enrichment of H3K4 methylation on the promoter of *DRAIC*, a long noncoding RNA known for its role in tumor suppression. In glioma cells, SET7/9 expression was found to be significantly reduced, leading to a corresponding decrease in H3K4 methylation at the *DRAIC* promoter, enhancing metastatic potential [26].

SETMAR is an HMT involved in several DNA processes such as DNA repair through the non-homologous end joining pathway and regulation of gene expression [27,28]. The SET domain is fused to an inactive DNA transposase domain, but both domains are involved in different cellular processes [27,28]. Studies have implicated a role for SETMAR in breast and colon cancers as well as AML and GBM, but that role remains unknown [2,27,29]. The shorter isoform of SETMAR, known as S-SETMAR, can serve as a prognostic marker for GBM, as individuals with longer survival times exhibited higher levels of this isoform in perilesional brain tissue [30,31]. However, researchers demonstrating this also showed higher levels of S-SETMAR than the full-length isoform in GBM tissues. In another study, researchers found that a short isoform of SETMAR, called SETMAR-1200, could be involved in GBM biogenesis [31]. Such results demonstrate that SETMAR’s role in GBM is complex [29,31].

SUV39H1 is an HMT involved in the trimethylation of H3K9, which is associated with transcriptional repression and heterochromatin formation. This gene plays significant roles in various cancers and neurodegenerative diseases. In cancer, SUV39H1 is often overexpressed, leading to increased H3K9 trimethylation and the consequent silencing of tumor suppressor genes, which promotes tumor progression in cancers such as prostate, breast, colon, lung, and DMG [32]. In GBM, SUV39H1 has been shown to have altered expression [33], and its activity can affect crucial pathways in cancer cell growth and therapy response.

NSD1, NSD2, and NSD3 are HMTs that make up the nuclear receptor-binding SET domain family. By modifying H3K36, the chromatin structure is maintained and the expression of genes controlling cell division, apoptosis, DNA repair, and epithelial-mesenchymal transition (EMT) is regulated [34]. Mutations and gene expression changes in this domain family have been known to lead to many cancer types. Overexpression is generally shown to be oncogenic, although some HMTs in this family can act as tumor suppressors [28,35].

NSD1 overexpression has been shown to promote AML, multiple myeloma, and lung cancer development [28,35]. It is also known to be involved in cancer cell migration and invasion in patients with breast cancer, hepatocellular carcinoma, or DMG [35]. However, the role of NSD1 is dependent on the tissue and etiological context [28,34]. For example, nonsense mutations present in about 10% of patients with squamous cell carcinoma of the head and neck potentially lead to oncogenesis, whereas some of these patients with NSD1 mutations have a better prognosis [28]. NSD2 has been implicated as an oncogene for multiple cancer types. Specifically, it has been associated with tumor aggressiveness for breast, cervical, lung, kidney, head and neck, blood, colorectal, prostate, skin, and ovarian cancers as well as brain cancers such as DMG [35,36]. Regarding DMG, also known as diffuse intrinsic pontine glioma, Yu et al. found that the co-knockdown of NSD1/2 reduced cell proliferation and tumorigenesis in H3K27 methionine substituted-DMG cell lines. NSD1/2 and LEDGF/HDGF2 were found to facilitate H3K27 methionine substituted-DMG tumor growth in vivo. Furthermore, regarding H3K36me2 in DMG cells, NSD1 and NSD2 were found to be key writers. Ideally, the SET domains of NSD1/2 could be targeted with inhibitors to reduce tumor growth, but such efforts have proven unsuccessful on a nanomolar scale. Instead, new inhibitors targeting the unique features of NSD1/2, such as unwrapping of linker and nucleosomal DNA, should be studied [36].

SMYD2, an HMT that methylates H3K4 and H3K36, is known to be involved in proliferation, epithelial–mesenchymal transition, and invasion in many cancer types [37,38]. Regarding brain cancers, SMYD2 has been shown to correlate with poor prognosis for glioma and GBM. Yu et al. looked at deregulated epigenetic enzyme gene signatures in LGGs to identify prognostic markers. They found that *SMYD2* was one of 13 aberrantly expressed genes in LGGs, and that SMYD2 was essential for cell proliferation and tumor formation. Additionally, SMYD2 silencing decreased the expression of other genetic regulators in an identified 13 gene-set, demonstrating that SYMD2 expression contributes to other elements involved in glioma formation [37]. In addition, Li et al. found that SMYD2 was significantly upregulated in GBM stem cells, that loss of function of SMYD2 decreased GBM cell proliferation, and that in in vivo assays, loss of SMYD2 stopped tumor growth. Monomethylation at the K224 residue of PGCla (PGCla K224me), which was elevated in GBM stem cells, was regulated by SMYD2, as high SMYD2 expression led to high PGCla K224me levels. Downstream effects of this pathway included inhibited mitochondrial biogenesis and increased growth of GBM stem cells [39]. However, further studies should be conducted to gain a deeper understanding of the role of SMYD2 in brain cancer, especially given SMYD2’s potential involvement in glioma and GBM therapy resistance. In an attempt to increase the sensitivity of GBM to radiotherapy, Li et al. found that targeting PGC1a K224me and nuclear transport of PGCla were promising [39]. Yu et al. found that both silencing and inhibiting SMYD2 in glioma cells stopped temozolomide resistance [37]. Pan et al. investigated SMYD2 and temozolomide resistance specifically. Not only did they find that increased SMYD2 content led to increased glioma progression and low survival probability but also inhibited glioma cell sensitivity to chemotherapy [38]. Therefore, downregulating SMYD2 may be a promising target for glioma and GBM treatment with the added benefit of enhancing chemotherapy and radiotherapy sensitivity.

DOT1L is an HMT that regulates transcription activation and elongation through H3K79 methylation. DOT1L is known to play critical roles during embryogenesis and normal development as it regulates stem and progenitor cell populations [40]. DOT1L has been implicated as an oncogene for MLL-rearranged leukemia, a finding that sparked the development of DOT1L inhibitors [2,40]. DOT1L has also been implicated to have a role in neuroblastoma and colorectal, ovarian, and breast cancers [40]. In GBM stem cell cultures, DOT1L was found to be more highly expressed than in the bulk tumor tissue. DOT1L regulated GBM cell proliferation and stemness through H3K79me2, and the inhibition of DOT1L decreased the H3K79me2 levels without affecting other histone marks [41]. Additionally, Li et al. identified DOT1L’s role in two major metabolic pathways in GBM cells. When reactive oxidation species accumulated in the GBM cells, lactate dehydrogenase A moved into the nuclei. The presence of nuclear lactate dehydrogenase A resulted in the activation of DOT1L, which hypermethylated the *BCAT1* gene and caused an elevated expression of it. Elevated BCAT1 methylation resulted in increased catabolism of branched-chain amino acids in GBM cells, leading to the production of glutamate. Glutamate was then used to synthesize thioredoxin, an antioxidant essential for maintaining redox state balance in tumor cells, promoting the progression of GBM [42].

PRDM6 is an HMT that primarily targets lysine 20 on histone H4 (H4K20me3), contributing to transcriptional repression and chromatin regulation. In cancer, PRDM6 plays a dual role that can be context-dependent. Researchers observed that PRDM6 can act as a tumor suppressor in some instances such as in breast cancer or can act as an oncogene such as in lung cancer and leukemia [43]. For example, in some cancers, PRDM6 expression is associated with favorable outcomes due to its role in silencing oncogenes and maintaining normal cellular functions. Conversely, PRDM6 may contribute to tumor progression in other contexts by affecting chromatin structure and gene expression patterns that promote malignancy. In a recent study, PRDM6 was found to drive oncogenesis in Group 4 MBs, and current research is being done to identify downstream targets [43]. PRDM6’s role is less well-defined in neurodegenerative diseases, but its involvement in chromatin modification in the brain suggests that it could impact neuronal gene regulation [44].

NO66 is an HDM that acts on H3K4 and H3K36. NO66 has been associated with poor prognosis for colorectal and prostate cancer. In human glioma tissues, NO66 has functions as an oncogene, as high expression of NO66 correlated with poor prognosis, cell proliferation, and advanced disease. Additionally, high levels of NO66 expression were associated with poor prognosis and induced cell proliferation in GBM tissue. Mechanistically, the knockdown of NO66 has been shown to reduce the expression of epidermal growth factor receptor and c-MYC protein, which are critical drivers of GBM development [45].

In addition to NO66, HDMs such as JARID1A play pivotal roles as oncogenes for GBM. Its involvement in tumor cell proliferation is particularly crucial for temozolomide-resistant GBM, a form of the disease that does not respond to standard chemotherapy. Inhibition of JARID1A has been shown to suppress the growth of this resistant subtype, highlighting its role in driving resistance mechanisms [46]. Of note is that JARID1A, while exhibiting the qualities of an oncogene, can act as a tumor suppressor, as studies have demonstrated that it supports cellular senescence by mediating retinoblastoma-interacting proteins [47]. More research of this family of modifiers is needed, as JARID1C, a related histone lysine modifier, has yet to be explored in relation to brain tumorigenesis.

LSD1, an HDM, removes a methyl group from mono- and dimethyl lysine on the H3K4 mark. It is overexpressed in various cancers such as breast cancer, gastric cancer, prostate cancer, hepatocellular carcinoma, esophageal squamous cell carcinoma, and AML [48]. Overexpression of this modifier leads to a decrease in cell differentiation and increase in tumor cell proliferation, indicating an oncogenic effect. Its expression also correlates with the proliferation of many neuronal cancers. LSD1 acts as an oncogene in Gfi1-activated MB. Thus, the inhibition of LSD1 suppresses tumor growth and significantly increases survival rates. Gfi1, a transcription factor driving an aggressive subtype of MB, is dependent on LSD1 activity for its oncogenic function. Blocking LSD1 activity disrupts Gfi1-mediated transcription, thereby reducing tumor proliferation [49]. LSD1 also exerts oncogenic effects on GBM by maintaining the ATF4-dependent integrated stress response, a protective pathway that enables tumor cells to survive under adverse conditions such as hypoxia and nutrient deprivation. By stabilizing this pathway, LSD1 ensures the efficient function of cellular mechanisms that mitigate stress. Inhibition of LSD1 leads to disruption of the ATF4-dependent integrated stress response, impairing the tumor cells’ ability to adapt to environmental stressors. This dysregulation sensitizes tumor cells, increasing their vulnerability to external stress and ultimately leading to cell death. Therefore, loss of LSD1 function compromises tumor resilience, reinforcing this HDM’s oncogenic nature [49].

JMJD5, part of the JumonjiC domain containing protein family, is an HDM that has been shown to promote the migration and invasion of and inhibit apoptosis in oral squamous cell carcinoma. It is also known to be involved in the progression of breast, lung, stomach, and liver cancer, although its expression differs for these cancer types [50,51]. Song et al. showed that JMJD5 expression was significantly elevated in GBM cells and tissue and that this was correlated with the pathological grade of glioma [50]. After JMJD5 knockdown, the pyruvate kinase enzymatic activity in GBM cells increased, whereas the glycolysis and proliferation rates decreased. JMJD5 overexpression produced the opposite results. With re-expression of JMJD5 in GBM cells, pyruvate kinase enzymatic activity also increased. This rescue of JMJD5 expression was mediated by HEATR5B-881aa, a protein from circular RNA that was later shown to reduce JMJD5 stability through phosphorylation. Additionally, HEATR5B-881aa overexpression inhibited GBM xenograft growth and prolonged the survival times of nude mice with GBM. This study revealed a novel mechanism of the regulation of aerobic glycolysis and proliferation in GBM cells through the ZCRB1/circHEATR5B/HEATR5B-881aa/JMJD5/PKM2 pathway, which can provide novel strategies and potential targets for GBM therapy [50].

Understanding the function of these HMTs and HDMs in patients with brain cancer is essential for developing therapies for it. Such research may also have implications for neurodegenerative diseases for multiple reasons. Both brain cancers and neurodegenerative diseases are associated with specific mutations, and genetic correlations in one disease could lead to findings for the other. Given that brain cancer results from uninhibited proliferation whereas AD, PD, and HD are caused by neuron degeneration, these diseases may have similar pathways with opposing mechanisms [52]. Additionally, some brain tumor treatments have been known to have degenerative effects on neurons. For example, researchers have shown chemotherapy to be a potential risk factor for AD [53]. Therefore, studies of brain cancer and neurodegenerative diseases could provide insights for multiple neuronal pathologies.

## 4. Roles of HMTs and HDMs in Neurodegenerative Diseases

In addition to brain cancer, HMTs and HDMs are involved in neuronal processes but can negatively impact brain function when mutated or amplified. Such effects can lead to neurodegenerative diseases such as AD, PD, and HD (Figure 4). Changes in histone methylation have been observed in AD, particularly affecting brain integrity and memory. In fact, increased trimethylation of H3K9 has been seen in the postmortem brains of patients with AD [54]. Regarding PD, the loss of dopaminergic neurons in patients with it is thought to result from a combination of genetic and environmental factors. Previous findings suggest that changes in histone methylation may control substantia nigra neurons in patients with PD [55]. Caused by mutations of the *HTT* gene, HD progression in its early stages is potentially caused by transcriptional dysregulation due to changes in histone methylation [56].

PRDM3 is a significant regulator of histone modifications, particularly those influencing the H3K9 methylation mark, associated with transcriptional repression. PRDM3 is an HMT that catalyzes the addition of methyl groups to H3K9, contributing to heterochromatin formation and gene expression silencing. Dysregulation of PRDM3 has been implicated to be involved in various pathological conditions. In cancer, low PRDM3 expression and PRDM3 mutations can disrupt normal gene silencing and contribute to tumorigenesis, as observed in many leukemias and hepatocellular and ovarian carcinomas in which an alteration in its function negatively impacts tumor suppressor genes and promotes oncogenic processes [57]. Regarding neurodegenerative diseases, PRDM3 could affect neuron differentiation because of its role in regulating the expression of genes crucial for neuronal development and function [44].

SUV4-20H2 is an HMT that specifically trimethylates H4K20, which, although not directly associated with H3K9, plays a role in chromatin structure maintenance and gene regulation. SUV4-20H2 is involved in maintaining heterochromatin stability and regulating transcriptional silencing. In cancer, its role is complex, and the overexpression of SUV4-20H2 can contribute to the progression of various cancers by enhancing the formation of heterochromatin and silencing tumor suppressor genes, which facilitates tumor growth and metastasis [58]. For instance, low SUV4-20H2 levels were present in lung adenocarcinoma and lung squamous cell carcinoma. However, SUV4-20H2 was amplified in pancreatic and kidney cancers, demonstrating that this enzyme has cancer type-dependent effects. In neurodegenerative diseases, the dysregulation of SUV4-20H2 may impact neuronal health by altering chromatin modifications that affect gene expression related to neuroprotection and neuronal survival [59].

ASH1L is an H3K4 and H3K36 HMT amplified in many cancers including breast, uterine, pancreatic, liver, and thyroid cancer and AML [28,35]. In prostate cancer, the overexpression of the C-terminal core of ASH1L increased the H3K36me2 levels, whereas the deletion of the domain had no effect. This difference in expression emphasizes the oncogenic behavior of ASH1L as it promotes cell cycle deregulation [60]. Although studies have confirmed ASH1L’s oncogenic role in cancer, research demonstrating its involvement in AD, PD, or HD is lacking. However, ASH1L has a potential association with dementia with Lewy bodies, which has a 31% genetic overlap with AD and PD. Using a large European sample, a genome-wide association study demonstrated that ASH1L was the closest gene to a dementia with Lewy bodies-related locus on chromosome 1 [61].

The HDM KDM7A plays a crucial role in the regulation of histone methylation by specifically targeting H4K20. By removing methylation marks from H4K20, KDM7A contributes to chromatin remodeling and gene expression modulation. Dysregulation of KDM7A has been implicated to have a role in various malignancies [62]. In prostate cancer, altered KDM7A function negatively affects genomic stability and contributes to tumor progression [63]. In neurodegenerative diseases, KDM7A disruption can impair neuronal gene expression and contribute to cognitive decline, which is often seen in AD. This enzyme’s role in modulating chromatin structure negatively affects genes involved in neuronal survival and function, exacerbating neurodegenerative processes [64].

### 4.1. HMTs and HDMs in AD

PRDM16 is an important HMT involved in the regulation of the H3K9 methylation mark, which is associated with transcriptional repression and chromatin compaction. PRDM16 functions as a tumor suppressor that adds methyl groups to H3K9, thereby contributing to gene silencing and heterochromatin formation. Dysregulation of PRDM16 has been implicated to play a role in several disease contexts. In cancer, alterations in PRDM16 expression or function have been associated with various leukemias as aberrant PRDM16 activity can disrupt normal gene expression and contribute to oncogenesis [57]. Additionally, PRDM16 mutations and altered PRDM16 expression have been linked to neurodegenerative diseases highlighting its role in neural stem cell maintenance and differentiation. Repression of PRDM16’s transcriptional activity due to sumoylation could impact AD pathogenesis as sumoylation plays a role in AD [44].

PRDM2, also known as RIZ, is an HMT involved in the trimethylation of lysine 9 on histone H3 (H3K9me3), a mark associated with transcriptional repression and heterochromatin formation. In cancer, PRDM2 often acts as a tumor suppressor; loss or reduced expression of it can lead to the dysregulation of H3K9me3, contributing to the activation of oncogenes and the silencing of tumor suppressor genes. This dysregulation is implicated to be involved in various cancers including leukemia and solid tumors as it can promote tumorigenesis and negatively impact prognosis [65]. Although PRDM2 has not been implicated to play a role in brain cancer, it does play a part in neurodegenerative diseases. Specifically, PRDM2’s role is crucial to maintaining neuronal function. Altered PRDM2 activity can disrupt neuronal gene expression and contribute to disease mechanisms for conditions like AD [44]. For example, Zhang et al. demonstrated that DNA breaks were upregulated in the gene encoding PRDM2 in AD brains, which could contribute to neurodegeneration [66].

The HDM JMJD1A is a key enzyme involved in the modulation of histone methylation, specifically by targeting H3K9. JMJD1A functions by removing repressive trimethylation marks from H3K9, thereby promoting an open chromatin structure and facilitating gene expression. JMJD1A has been implicated to have a role in various malignancies including prostate and breast cancers. Its overexpression can drive tumorigenesis by upregulating oncogenes and enhancing cell proliferation. For instance, in prostate cancer, JMJD1A’s activity is linked to the activation of genes that support tumor growth and progression. In breast cancer, JMJD1A affects estrogen receptor signaling pathways, contributing to cancer cell proliferation [67]. In neurodegenerative diseases, the dysregulation of JMJD1A can disrupt normal neuronal gene expression. When studying the effects of circadian rhythm disruption on AD, Li et al. found that JMJD1A played a protective role in the response to neuroinflammation [68]. Consequently, JMJD1A’s impact on histone methylation and chromatin dynamics highlights its potential as a therapeutic target for both cancer and neurodegenerative conditions, offering avenues for interventions aimed at restoring normal gene expression and cellular function.

SETDB2 is an HMT specifically targeting H3K9me3, contributing to transcriptional repression and heterochromatin formation. In the context of cancer, SETDB2 is often implicated to be involved in tumorigenesis; its overexpression can lead to abnormal H3K9 trimethylation, which results in the silencing of tumor suppressor genes and the promotion of cancer cell proliferation and survival. For instance, SETDB2 has been linked to poor prognosis for a specific AML subtype and has been observed to influence the progression of other malignancies, such as renal and gastric cancer, by altering chromatin structure and gene expression [69]. SETDB2’s role is also significant in neurodegenerative diseases as it negatively impacts the regulation of genes involved in neuronal function and stress responses. Abnormal SETDB2 activity can affect the expression of genes related to neuroinflammation and neuronal cell death, contributing to disease pathology in conditions like AD [70].

### 4.2. HMTs and HDMs in AD and PD

NSD3 overexpression has been associated with many cancers such as lung squamous cell carcinoma, squamous cell carcinoma of the head and neck, and AML as well as cancers of the bone, breast, colorectum, and pancreas [28,35]. Because of this overexpression, NSD3 is categorized as an oncogene, although its role in tumor progression remains unclear. On top of its role as an oncogene in many diverse cancers, NSD3 is highly expressed in the brain and has been associated with AD and PD. Sarnowski et al. found that whole-exome sequencing data demonstrated that NSD3 was significantly associated with circulating total tau levels when aggregating rare variants of NSD3 with high or moderate impact on protein function. This is significant as circulating tau protein levels are known to be associated with AD and two PD subtypes. These results included missense and loss-of-function variants of NSD3, but rare variants mostly contributed to NSD3 associations with circulating tau levels [71]. More studies are needed to understand NSD3’s potential causal role in these tauopathies.

PHF2 is an HDM that plays a crucial role in epigenetic regulation by modulating histone methylation, mainly targeting H3K9. By removing repressive trimethylation marks from K9, PHF2 promotes a more open chromatin structure, thereby influencing gene expression and cellular processes. Deregulation of PHF2 has been linked with various malignancies, such as renal cell carcinoma and breast cancer, as abnormal PHF2 activity disrupts the expression of oncogenes and tumor suppressor genes, facilitating tumor growth and progression [72]. In neurodegenerative diseases, such as AD and PD, dysregulated PHF2 activity impacts neuronal gene expression, contributing to cognitive decline and DNA damage [11,73].

JMJD3 encodes an HDM that plays a pivotal role in regulation of the H3K27 methylation mark, a modification associated with transcriptional repression by PRC2. JMJD3 specifically targets and removes the repressive H3K27me3 mark, thus facilitating gene activation and influencing various cellular processes. Aberrations in JMJD3 activity have been implicated to be involved in several diseases. JMJD3 is often upregulated in various malignancies, such as lung cancer and breast cancer, in which increased JMJD3 activity can contribute to oncogene activation and tumor progression [74]. Additionally, JMJD3 mutations and dysregulation have been associated with neurodegenerative diseases due to its critical role in neuronal gene expression and development. The gene is commonly implied to be an epigenetic regulator in both AD and PD [75,76].

### 4.3. HDMs Involved in HD

UTX is a crucial component of the histone modification landscape, particularly influencing the H3K27 mark, which is associated with transcriptional repression. UTX functions as an HDM that specifically removes the H3K27me3 mark, counteracting PRC2-mediated repression. Dysregulation of UTX has been implicated to be involved in several cancers including renal cell carcinoma and multiple myeloma, as loss or mutation of UTX can lead to aberrant gene silencing and tumorigenesis [77]. UTX mutations have been associated with neurodegeneration and HD due to its role in maintaining the proper gene expression patterns necessary for neuronal function [56]. Thus, UTX serves as a critical regulator of gene expression with significant implications for both cancer and neurodegenerative diseases.

Whereas the specific modifiers described above play roles in neurodegenerative diseases, they are not involved in brain cancers. However, brain cancer findings are relevant to neurodegenerative diseases due to two processes: neuron crosstalk and neuron differentiation. For example, many studies have demonstrated that molecules involved in both brain cancer and neurodegenerative diseases have shared roles. Multiple signaling pathways that regulate cell death and survival in cancer have also been correlated with neurodegenerative diseases. Such pathways involve DNA damage, cell cycle aberrations, oxidative stress, inflammation, and immunity [52]. Additionally, in the early stages of neuron development, defects in modifiers that play crucial roles in differentiation could cause cancer or neurodegenerative diseases. For example, KMT2D has been shown to induce the expression of many genes involved in differentiation. The heterozygous loss of KMT2D downregulated cerebellar granule cell differentiation and induced MB [78]. While neurodegenerative diseases do not usually arise from differentiation issues, there are molecular pathways involved in maintaining cell differentiation. Signaling pathways such as Wnt and Notch have been known to be disrupted in many neurodegenerative diseases as well as cancer [79]. Therefore, investigation into either pathology could have implications for both.

## 5. Roles of HMTs and HDMs Implicated in Both Brain Cancer and Neurodegenerative Diseases

Furthermore, the roles of histone lysine modifiers in both neurodegenerative diseases and brain cancer have been explored (Appendix A). Some HMTs and HDMs have only been shown to be mutated in these pathologies, whereas other modifiers have been documented to induce methylation changes, and downstream genes and pathways have been identified. Understanding the interconnections and divergences between these diseases will reveal potential targets for novel brain cancer and neurodegenerative disease therapies (Figure 5).

### 5.1. Role of SETD2 in Brain Cancer and Neurodegenerative Diseases

SETD2 is an HMT that trimethylates H3K36 and regulates DNA methylation, DNA repair, RNA processing, and tumor suppression. It has been implicated to have a role in lung adenocarcinoma, multiple leukemias, hematological malignancies, bladder cancer, gastrointestinal cancers, and central nervous system tumors [28]. This HMT is known to be a tumor suppressor, as *SETD2* mutations are involved in progression, recurrence, and survival outcomes for such cancers. *SETD2* truncation mutations account for more than 30% of pediatric HGGs [35]. *SETD2* mutations are found in many central nervous system tumors including LGGs, HGGs, and non-glial tumors in pediatric patients, young adults, and patients older than 55 years. Mutations of *SETD2* in primary occurring HGGs include frameshifts, truncations, and point mutations at high allele frequencies, whereas those in recurrent HGGs include *SETD2* point mutations at lower allele frequencies [80]. Multiple studies have implicated the involvement of SETD2 in IDH1/2 wild-type (IDH-WT) GBM, a subtype that lacks many GBM characteristics like EGFR amplification. A ZDHHC16/SETD/H3K36me3 signaling axis was shown to be inactivated in EGFR-altered GBM, which reduced palmitoylation [81]. Additionally, multiplatform molecular analysis showed that *SETD2* was a frequent inactivating mutation in IDH-WT GBM [82]. Based on gene signature analyses, *SETD2* mutations were correlated with poor prognosis for IDH-WT GBM patients. Through transforming growth factor (TGF)-β1 signaling, mutated *SETD2* tumor cells were found to activate HGG-associated microglia in IDH-WT GBM, which promoted tumor progression. Regarding neurodegenerative diseases, SETD2 has been shown to be involved in the mechanism promoting HD [83]. Gao et al. found that Htt yeast two-hybrid protein B (HYPB), also known as SETD2, directly interacted with Htt and was involved in HD pathology. Polyglutamine-expanded Htt isolated. Understanding the molecular interaction between HYPB/SETD2 and mutant Htt could have implications for the pathology of HD [84]. In both brain cancer and HD, decreases in methylation due to either inactivating mutations or cytoplasmic localization contributed to the promotion of these pathologies.

### 5.2. Roles of G9a and GLP in Brain Cancer and Neurodegenerative Diseases

G9a, also known as EHMT2, and GLP, also known as EHMT1 and G9a-like protein, are HMTs that act on H3K9. GLP has been found to be elevated in a few cancers such as breast, lung, gastric, and ovarian cancers whereas G9a is frequently overexpressed in many cancers including breast, lung, ovarian, prostate, colon, bladder, and hepatocellular cancers [85]. Both of these HMTs have been found to have abnormalities in brain cancer. However, G9a’s role in brain tumors is better understood. In a pediatric case of metastatic ganglioglioma, a duplication breakpoint was present within intron 16 of *GLP*, which resulted in a truncation [86]. This study suggested that GLP functions as a tumor suppressor, demonstrating that it has cancer-type-dependent effects. Additionally, crosstalk between HMTs and HDMs can regulate *RE1* silencing transcription factor (*REST*) gene expression, which is abnormally elevated in SHH MBs. Researchers have demonstrated that during the proliferation of cerebellar granular progenitor cells, the origin of this MB subgroup, PRC2 and G9a/GLP complex acted in a repressive manner, contrary to elevated KMT2D and KDM7A, which maintained *Rest* homeostasis. Expression of G9a was elevated in proliferating cerebellar granular progenitor cells whereas differentiating cells had low levels of G9a [87]. This suggests that the dysregulation of G9a impacts neurodifferentiation and could influence the proliferation of SHH MBs. G9a’s role in MB was further characterized by Souza et al., who found higher levels of G9a in SHH, Group 3, and Group 4 MB tumors than in Wnt tumors. Of note, increased transcription of the *G9a* gene correlated with shortened survival in patients with SHH MB. Dose-dependent inhibition of G9a decreased MB cell viability, demonstrating G9a’s oncogenic nature [88].

G9a has also been implicated to have a role in GBM, as it has been found to promote tumor growth and decrease survival times in a mouse model. Cao et al. identified the mechanism by which G9a induces tumorigenesis in that it bound to Fbxw7, a Notch suppressor, resulting in increased H3K9me2 of the Fbxw7 promoter and inhibition of the transcription of immune-associated molecules. G9a upregulated programmed death-ligand 1, an immune checkpoint molecule, and downregulated MHC-I on the surface of tumor cells through the Notch pathway, which resulted in an inhibited immune response in glioma stem cells. Additionally, G9a was suggested to promote the proliferation and stemness characteristics of glioma stem cells and inhibit apoptosis [89]. In these studies, G9a was shown to engage in crosstalk with other histone lysine modifiers and upregulate the Notch signaling pathway, demonstrating an oncogenic effect on brain tumors.

In neurodegenerative diseases, the roles of G9a and GLP are also critical, as their dysregulation can impact neuronal gene expression and contribute to disease pathology. In progressive HD mouse models, G9a and GLP were found to be mutated, and these mutations overlapped with transcription decreases. The downregulation of epigenetic-related genes was highly associated with the downregulation of dopaminergic neuronal markers in these mice [90].

Whereas the downregulation of G9a and GLP was associated with HD, the opposite is true regarding their roles in PD and AD. Zhang et al. found a link between H3K9 dimethylation and the pathology of α-synuclein, which aggregates intracellularly in PD. Used to study PD progression, treatment with α-synuclein performed fibrils in mice upregulated G9a and GLP and increased H3K9 dimethylation. Treatment with α-synuclein performed fibrils also increased H3K9 dimethylation at the promoters of the synaptic genes *SNAP25*, *PSD95*, *Synapsin 1*, and *vGLUT1*, which downregulated the synaptic-related protein expression. When G9a and GLP were inhibited with A-366 or short hairpin RNA, the synaptic protein expression was restored in primary neurons, but the performed fibril-induced formation of α-synuclein was not impacted. Such findings suggest that G9a and GLP methylation enhancements are likely downstream effects of α-synuclein performed fibril [91]. Similarly, Zheng et al. found that H3K9 dimethylation was significantly elevated in the hippocampus and prefrontal cortex in a late-stage familial AD mouse model and in the prefrontal cortex tissues obtained postmortem from a patient with AD. In the prefrontal cortex and hippocampus of mice with familial AD, elevated H3K9 dimethylation correlated with repressed glutamate receptor transcription, and the prefrontal cortex G9a and GLP levels were significantly elevated. Treatment with G9a and GLP inhibitors reversed this methylation, leading to the recovery of glutamate receptor expression and function in the prefrontal cortex and hippocampus of the mice. These results suggest that the loss of glutamate receptors in AD results from increased methylation by G9a and GLP. This methylation was found to occur at the NDMA and AMPA receptor genes *Grin2a*/NR2A, *Grin2b*/NR2B, and *Gria2*/GluA2 as well as many genes related to cell communications, signal transduction, neurological system processes, G-protein coupled receptor signaling, and sensory perception. Upon treatment with the G9a and GLP inhibitor BIX01294, the methylation of many genes in these categories was reversed, although amyloid-β plaques and tau protein levels were not decreased by treatment with G9a and GLP inhibitors [92].

In summary, G9a and GLP upregulation contributed to the downregulation of synaptic proteins and glutamate receptors in PD and AD mouse models, respectively. Whereas G9a and GLP inhibition did not impact the pathogenesis of these diseases, it did rescue motor impairments in PD mice and restore recognition, working, and spatial memory in AD mice [91,92]. Therefore, G9a and GLP inhibition could still function as effective therapy by targeting elevated H3K9 levels in PD and AD.

### 5.3. Roles of KDM2A and KDM2B in Brain Cancer and Neurodegenerative Diseases

KDM2A, also known as JHDM1A and FBXL11, is an HDM that plays a role in cell proliferation, differentiation, senescence, and apoptosis by regulating H3K36 methylation. KDM2A is shown to be oncogenic in many cancer types including breast, gastric, and lung cancers [93]. KDM2A is known to play a role in the development of GBM in two different pathways, specifically through microRNAs (miRNAs), which are essential regulators of gene expression in human GBM and other human cancer types. Two studies have demonstrated that the miRNAs miR-3666 and miR-663a were significantly downregulated in GBM tissues and cells, which correlated with poor prognosis. MiR-3666 and miR-663a both targeted KDM2A, and the upregulation of these miRNAs decreased the expression of KDM2A in the GBM cells [93,94]. Mechanistically, Wang et al. found that knocking down KDM2A attenuated the TGF-β/Smad pathway, which is essential for the progression and invasiveness of GBM cells. KDM2A knockdown also inhibited TGF-β and the phosphorylation of TGF-β downstream effectors SMAD1 and SMAD2. Therefore, when miR-663a was downregulated in the GBM cells, KDM2A was upregulated, increasing the phosphorylation of SMAD1 and SMAD2. This phosphorylation is essential for the translocation of SMAD1 and SMAD2 to the nucleus, where they impact the expression of genes involved in cell differentiation and apoptosis [94].

Another miRNA implicated in brain cancer is miR-302a, as it was downregulated in glioma samples. Long noncoding RNA homeobox A cluster antisense RNA 2 (lncRNA HOXA-AS2), which has been found to be elevated in glioma tissues, upregulated KDM2A by inhibiting miR-302a’s ability to bind to the 3′ untranslated region of KDM2A. The upregulation of KDM2A contributed to regulatory T-cell proliferation and immune tolerance in glioma samples and a tumor xenograft mouse model as it demethylated the jagged 1 (JAG1) promoter region, elevating JAG1 expression and contributing to regulatory T-cell proliferation. In the Notch signaling pathway, JAG1 promotes tumorigenesis by inducing angiogenesis and immune cell infiltration. Overall, Zhong et al. found that glioma cell proliferation and immune tolerance resulted from the inhibition of miR-302a through lncRNA HOXA-AS2, which upregulated the KDM2A/JAG1 axis [95].

Regarding potential therapies for glioma and GBM, upregulating miRNAs could reduce stemness and progression, as the overexpression of miR-663a inhibited the KDM2A/TGF-β/Smad signaling pathway [94]. Downregulation of KDM2A is also a promising target for glioma and GBM treatment. Additionally, more research could be conducted to confirm the downstream genetic targets of the KDM2A/TGF-β/Smad pathway and the KDM2A/JAG1 axis in GBM and glioma to improve treatments and mitigate their secondary effects in patients.

KDM2B, also known as JHDM1B and FBXL10, is an HDM that modifies H3K36 and regulates cell growth, the cell cycle, and tumorigenesis [96]. It has been shown to be an oncogene in AML, pancreatic cancer, synovial sarcoma, glioma, and GBM by maintaining stem cell properties, inhibiting differentiation pathways, and activating oncogenic pathways [97]. KDM2B also regulates the self-renewal capacity of cancer and cancer stem cells in hematological and pancreatic cancers. Its expression is also positively correlated with advanced cancer stages [98].

KDM2B is highly expressed in glioma and GBM cells, and many studies have elucidated downstream targets and pathways [96,99]. Staberg et al. established that KDM2B was required for GBM cell survival, as KDM2B loss led to the apoptosis and accumulation of DNA damage in GBM cells. This DNA damage impaired self-renewal and the maintenance of GBM stem-like cells, leading to a decrease in GBM stem cell frequency, differentiation capacity, and survival [99]. Additionally, Kurt et al. identified KDM2B as a novel regulator of tumor necrosis factor-related apoptosis-inducing ligand (TRAIL) response, which works to kill tumor cells selectively. Silencing of KDM2B enhanced caspase-8, -3, and -7 activation, poly(ADP-ribose) polymerase cleavage, and the sensitivity of GBM cells to TRAIL, which all play a role in apoptosis. KDM2B knockdown accelerated apoptosis by upregulating proapoptotic genes such as Harakiri (*HRK*), *caspase-7*, and death receptor 4 (*DR4*). Loss of KDM2B also suppressed antiapoptotic genes and inhibited tumor growth in vivo [98]. These results suggest that KDM2B is an epigenetic regulator of glioma migration and proliferation as well as GBM cell apoptosis, making it a promising target for future glioma and GBM treatment.

Wang et al. found that high levels of KDM2B expression correlated with high glioma grades and low survival rates. In glioma cell lines, silencing of KDM2B increased the expression of P21 and decreased the expression of cyclin D1, a pro-oncogene. Elevated P21 and diminished cyclin D1 would inhibit the cell cycle by repressing retinoblastoma phosphorylation and decreasing the G1 to S transition, respectively. Additionally, KDM2B knockdown decreased the expression of KMT6A, an HMT known to promote the migration and progression of tumors. Researchers showed that KDM2B inhibits the transcription of miR-let-7b, which upregulates KMT6A. However, more research should be conducted to identify the downstream genetic targets of this pathway [95]. KDM2B’s upregulation of the cell cycle in glioma also demonstrates a potential point of connection with AD, PD, and HD as cell cycle aberrations contribute to the development of neurodegenerative diseases.

Regarding neurodegenerative diseases, the HDMs KDM2A and KDM2B were shown to play a role in AD. Using postmortem brain tissues obtained from patients with AD, KDM2A expression was shown to be markedly higher than the non-demented controls. However, KDM2B mRNA was downregulated in the brains of patients with AD. In *Drosophilia*, knockdown of *kdm2*, a homolog of human *KDM2*, improved tauR406W-induced defects and locomotion impairment. These results suggest that KDM2 may regulate neuronal function, and neuronal deficiencies of KDM2 could exacerbate tauR406W-induced locomotion impairment [100]. These findings demonstrate that targeting KDM2A and KDM2B dysregulation in the brain could function as a potential AD therapy by relieving locomotion deficits. However, more research should be undertaken to identify the downstream genetic targets of KDM2A and KDM2B in AD, especially considering the contrary regulation of these modifiers observed in postmortem brain tissue from patients with AD.

Overall, our findings demonstrate that specific HMTs and HDMs play a role in both brain cancer and neurodegenerative diseases. These findings also reveal the involvement of these modifiers in neuron crosstalk and neuron differentiation pathways, which have relevance for both brain cancer and neurodegenerative diseases [52,78]. Regarding crosstalk, G9a, GLP, and KDM2B interacted with and influenced other HMTs and HDMs in three types of brain cancer to enhance demethylation. Identifying the downstream targets of these pathways would be interesting, especially given that the dysregulation of G9a and GLP in AD mice functioned to enhance methylation, and KDM2B, a demethylase, was downregulated in postmortem brain tissue from patients with AD. Additionally, multiple KDM2A and KDM2B studies have demonstrated that miR-663a, miR-302a, and miR-let-7b downregulation resulted in the tumorigenesis of GBM and glioma [94,95,96]. These findings are relevant to neurodegenerative diseases given that miRNAs play a role in such pathologies. In fact, miR-663a was found to be involved in neurodifferentiation in AD, although their effect on histone methylation is unknown [101]. Furthermore, the downstream effects of these miRNAs on GBM and glioma included the upregulation of Notch signaling pathways, which have been shown to affect neurodifferentiation in neurodegenerative diseases [79]. Therefore, an essential step is to further elucidate the downstream effects of HMT and HDM crosstalk as well as neurodifferentiation through miRNAs and Notch signaling pathways in brain cancer and neurodegenerative diseases to advance the treatment of these pathologies.

## 6. Conclusions and Future Directions

Many histone lysine HMTs and HDMs are known to have oncogenic, tumor suppressive, or cancer type-dependent effects. On top of its correlation with poor prognosis in LGGs, HGGs, GBM, and MB, the dysregulation of HMTs and HDMs in the brain has been shown to enhance cancer cell stemness and proliferation. Some histone lysine modifiers also inhibit brain cancer sensitivity to chemotherapy and radiation therapy. Additionally, changes in histone methylation and transcriptional regulation have been shown to contribute to neurodegenerative diseases. When modifiers essential for neuronal function are overexpressed or inhibited, abnormal gene regulation occurs, leading to neuroinflammation, cell death, and cognitive decline in patients with AD, PD, or HD. Due to neuron crosstalk and the role of cell differentiation in neurodegenerative diseases and brain cancer, studies of histone modifiers in both pathologies are significant. However, few modifiers are involved with both brain cancers and neurodegenerative diseases, highlighting the need for more research.

A future study of KDM2B and PRC2 could be interesting, as they have been shown to interact with each other [97]. Given the role of KDM2B and PRC2 as oncogenes in brain cancer, further research should be carried out to determine whether the upregulation of KDM2B and PRC2 in glioma, GBM, and MB plays a combinatory role in the self-renewal properties of these pathologies. Dysregulation of PRC2 and KDM2B has been shown to promote and repress AD, respectively. Therefore, more research may identify how alterations of KMT2B and PRC2 in brain cancer and its treatment could potentially trigger AD development.

Despite the documented roles of HMTs and HDMs in neurodegenerative diseases and brain cancer, limited treatment options are available. Although treatments do exist for AD, PD, and HD, they mainly focus on managing symptoms, such as musculoskeletal pain and neurobehavioral issues, and no curative treatments are available [8,10,54]. For oncogenic histone lysine modifiers, inhibitors would ideally diminish cancerous effects, but problems can arise, as these modifiers are essential for brain function. Treatment with the promising DOT1L inhibitor EPZ-5676 stopped the growth of GBM stem cells without effecting non-GBM cancer cell lines. After this treatment, self-renewal and cellular migration/invasion were reduced, and similar results were seen in GBM stem cells and immunodeficient mouse models [41]. However, EPZ-5676 and many other inhibitors targeting HMTs and HDMs in brain cancer are not approved for treatments in humans yet. Additionally, some histone lysine modifiers benefit from a combination with different drug classes, as diverse treatment types kill cells through different mechanisms. For example, combining chemotherapeutic agents like lomustine or etoposide with GSK-J4, a KDM2B inhibitor that decreases GBM cell viability and sensitizes GBM cells to chemotherapy, is a promising GBM treatment [99]. However, only one such drug combination is approved for brain cancer treatment [102]. Finally, some modifiers have been shown to contribute to brain cancer cell resistance to chemotherapy and radiation therapy, further highlighting the need for targeted, safe HMT- and HDM-based treatment options [37,38,39,47].

## Figures and Tables

**Figure 1 biology-13-01008-f001:**
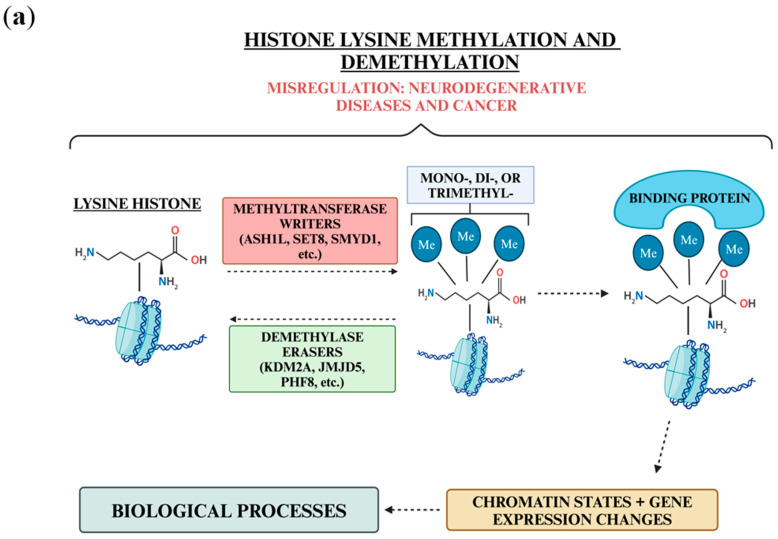
Effect of methylation and demethylation with associated histone lysine modifiers. (**a**) Overall mechanism of methylation and demethylation on the R group nitrogen of the histone lysine tail. Methylation and demethylation both affect a tighter or looser chromatin structure, leading to gene expression changes and impacting biological processes. (**b**) List of methyltransferases (M) and demethylases (D) acting on histone 3 or 4 lysine tails. Marks are shown as activators (green) or repressors (red) of gene expression. Both methyltransferases and demethylases are included under the specific marks they act on.

**Figure 2 biology-13-01008-f002:**
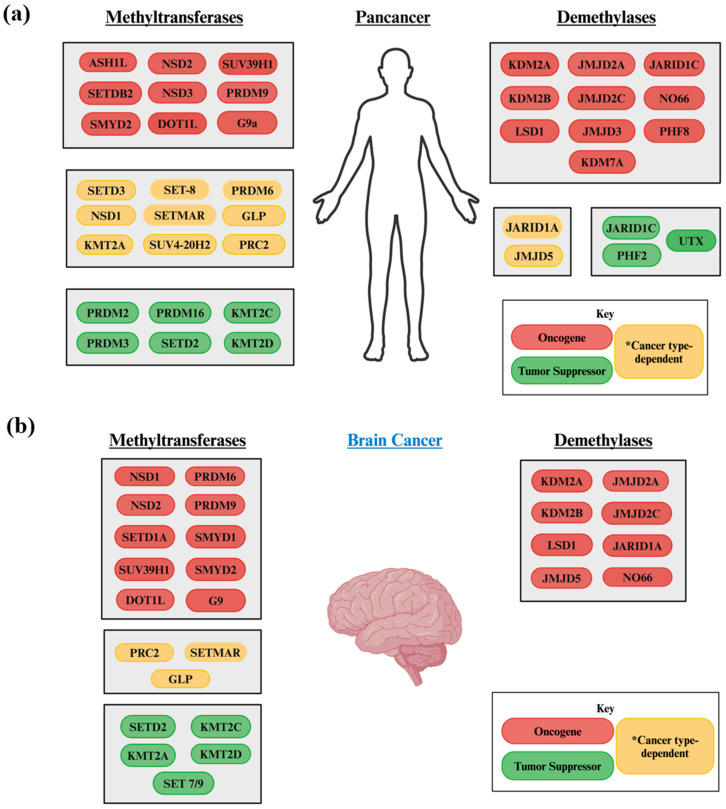
The HMTs and HDMs involved in multiple cancer types. HMTs and HDMs have been shown to affect the genesis and proliferation of (**a**) a wide range of cancers and (**b**) brain cancers. A pancancer designation was given to histone lysine modifiers affecting three or more cancer types. Each modifier is listed as having a documented oncogenic or tumor suppressive effect. Modifiers that act as oncogenes or tumor suppressors in different types of cancer are depicted as having a * cancer type-dependent effect.

**Figure 3 biology-13-01008-f003:**
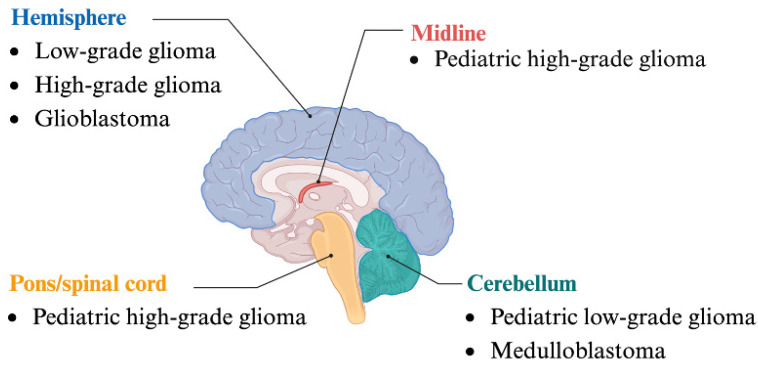
Specific types of brain cancer and the locations in the brain where they most frequently occur. Illustration of the brain showing highlighted regions where specific types of brain cancers, such as adult and pediatric low-grade and high-grade gliomas as well as glioblastoma and medulloblastoma, commonly occur. The locations depicted include the hemisphere, midline, pons/spinal cord, and cerebellum.

**Figure 4 biology-13-01008-f004:**
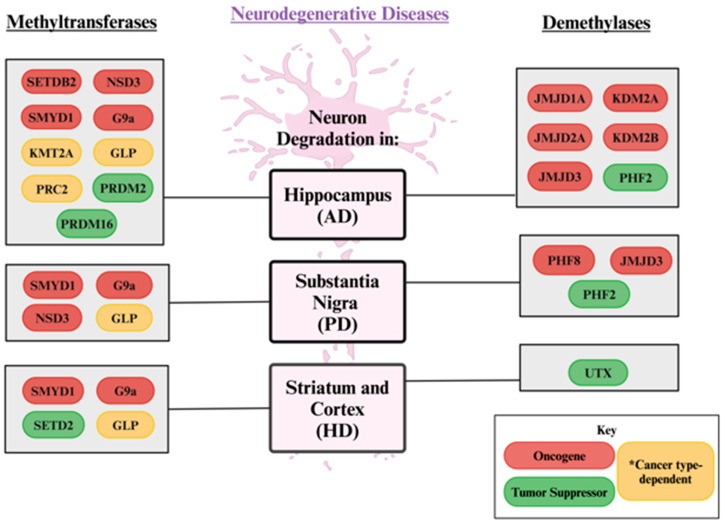
The HMTs and HDMs associated with Alzheimer’s disease (AD), Parkinson’s disease (PD), and Huntington’s disease (HD). These HMTs and HDMs have activating or inhibiting effects on neuron degradation in different brain structures. Neuron degradation in the hippocampus, substantia nigra, and striatum/cortex is known to contribute to AD, PD, and HD, respectively. Each modifier is listed as having a documented oncogenic or tumor suppressive effect. Modifiers that act as oncogenes or tumor suppressors in different types of cancer are depicted as having a * cancer type-dependent effect.

**Figure 5 biology-13-01008-f005:**
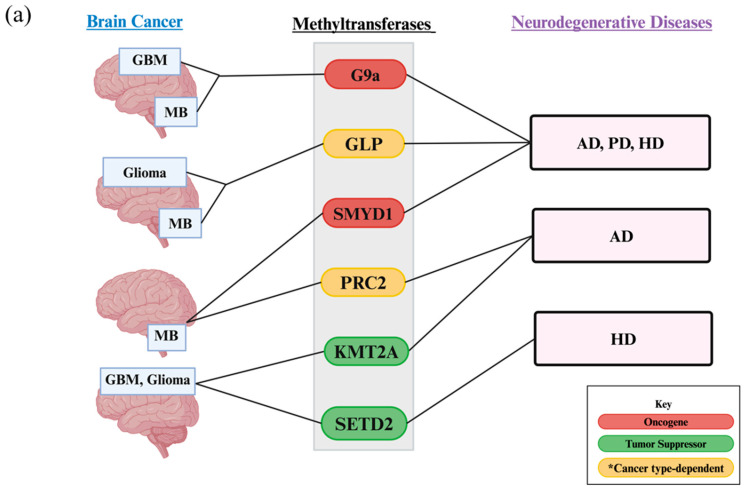
The (**a**) HMTs and (**b**) HDMs associated with both brain cancers and neurodegenerative diseases. Neuron degradation in the hippocampus, substantia nigra, and striatum/cortex is known to contribute to AD, PD, and HD, respectively. Dysregulation of histone lysine modifiers contributes to these neurodegenerative diseases. The impact of these modifiers on brain cancers such as medulloblastoma (MB), glioma, glioblastoma (GBM), and pediatric HGG (pedHGG) is also shown. Modifiers that act as oncogenes or tumor suppressors in different types of cancer are depicted as having a * cancer type-dependent effect.

## Data Availability

Not applicable.

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
