# Peer review of "Epigenetic Modifiers: Exploring the Roles of Histone Methyltransferases and Demethylases in Cancer and Neurodegeneration"

_biology, 2024, doi:10.3390/biology13121008_

Round 1
Reviewer 1 Report
Comments and Suggestions for Authors
Line 47: Indeed, histone lysine methylation is NOT a post-translational modification, since its effect is not on the histone expression itself. It affects other genes. Hence, this process is a pre-transcription modification. Please correct this point.
Line 84: Please clarify whether you mean that PD is a disease the most progressive neurological disease or it has the most growing incidence within population? Based on your reference the second option is correct. However, your reference is not a statistical and first-hand evidence. It is recommended to replace the reference with a more-suitable one.
Line 95: The statement needs a reference.
Line 110: The phrase “Many of these histone lysine modifiers”, according to the previous sentence implies like that H3K4, H3K36, and H3K79 are HMTs and HDMs, which is obviously wrong. Please correct the phrase accordingly to avoid misunderstanding.
Lines 162-166: Citation is needed. Also, some explanations about possible mechanisms and regulated genes are recommended.
All over the manuscript: Figures 1a, 1b, 3, 4, and 5 are cross-referenced within the manuscript, while there was no figure! Also, it seems that figure 2 has been forgotten in the order of numbering.
Lines 345-347: Citation is needed.
All over the manuscript: There is a significant lack of hierarchical order of contents. The paragraphs and content have been described with no link and it seems authors have been in a hurry to summarize a large amount of data. It is strongly recommended to speak about the exact types of mutations and their effect both on tumors and NDDs. Also, it is strongly recommended to speak about the statuses of downstream genes which are expressed or inhibited due to each methylation/demethylation. Thus, the manuscript can reach a more reliable and stronger conclusion. The tables are good, figures seem good. Only It is recommended to determine the genes which their methylation is changed during both cancers and NDDs. This determinations can be shown with a special sign or color.
Lines 713-715: The statement “However, few modifiers are involved with both brain cancers and neurodegenerative diseases, highlighting the need for more studies” needs more explanations and justifications, since the principal idea of the manuscript was based on the opposite of this statement.
Author Response
Thank you for your valuable comments. We greatly appreciate the insight you have provided to strengthen our manuscript.
Comments 1: Line 47: Indeed, histone lysine methylation is NOT a post-translational modification, since its effect is not on the histone expression itself. It affects other genes. Hence, this process is a pre-transcription modification. Please correct this point.
Response 1: Histone methylation is generally classified as a post-translational modification as it occurs after histones are initially translated (see Hyun and Li). However, we also understand that histone methylation can be looked at as a pre-transcriptional modification, as it can activate transcription of genes.
See:
Hyun, K.; Jeon, J.; Park, K.; Kim, J. Writing, erasing and reading histone lysine methylations. Exp Mol Med 2017, 49, e324, doi:10.1038/emm.2017.11.
Li, M.; Zhang, Z.; He, L.; Wang, X.; Yin, J.; Wang, X.; You, Y.; Qian, X.; Ge, X.; Shi, Z. SMYD2 induced PGC1alpha methylation promotes stemness maintenance of glioblastoma stem cells. Neuro Oncol 2024, 26, 1587-1601, doi:10.1093/neuonc/noae090.
Comments 2: Line 84: Please clarify whether you mean that PD is a disease the most progressive neurological disease or it has the most growing incidence within population? Based on your reference the second option is correct. However, your reference is not a statistical and first-hand evidence. It is recommended to replace the reference with a more-suitable one.
Response 2: Based on the reference in the manuscript, we thought that PD was growing the most in incidence. However, we identified current research demonstrating that PD is the second most common age-related neurodegenerative disorder diagnosed in North America, with an estimated 108 to 212 cases per 100,000 individuals aged 65 years and older. This rate decreases when including a broader group of individuals aged 45 and years and older, underscoring the increased prevalence of PD within the elderly population. Because of this finding, we have removed “the fastest growing” from the manuscript.
See:
Willis, A. W., Roberts, E., Beck, J. C., Fiske, B., Ross, W., Savica, R., Van Den Eeden, S. K., Tanner, C. M., Marras, C., & Parkinson’s Foundation P4 Group (2022). Incidence of Parkinson disease in North America. NPJ Parkinson's disease, 8(1), 170. https://doi.org/10.1038/s41531-022-00410-y
Comments 3: Line 95: The statement needs a reference.
Response 3: We have cited the reference below in the manuscript.
Park, J., Lee, K., Kim, K. et al. The role of histone modifications: from neurodevelopment to
neurodiseases. Sig Transduct Target Ther 7, 217 (2022). https://doi.org/10.1038/s41392-022-01078-9
Comments 4: Line 110: The phrase “Many of these histone lysine modifiers”, according to the previous sentence implies like that H3K4, H3K36, and H3K79 are HMTs and HDMs, which is obviously wrong. Please correct the phrase accordingly to avoid misunderstanding.
Response 4: We have corrected the sentence to say: “Many histone lysine modifiers affecting these marks were implicated across multiple cancer types.”
Comments 5: Lines 162-166: Citation is needed. Also, some explanations about possible mechanisms and regulated genes are recommended.
Response 5: We included lines 162-166 and 155-156 to transition from specific brain cancer locations to modifier discussions. Later in this section and in the rest of the manuscript, we further explained these modifiers’ mechanisms in desmoplastic medulloblastoma and diffuse midline glioma with necessary citations. However, we understand the concerns and removed these lines from the manuscript.
Comments 5: All over the manuscript: Figures 1a, 1b, 3, 4, and 5 are cross-referenced within the manuscript, while there was no figure! Also, it seems that figure 2 has been forgotten in the order of numbering.
Response 6: Figures were sent to reviewers by editors after the first review. Figure 2 was not forgotten in the order numbering and was cross-referenced at the end of Section 2.
Comments 7: Lines 345-347: Citation is needed.
Response 7: We have added the citation below into the manuscript.
Klokkaris A, Migdalska-Richards A. An Overview of Epigenetic Changes in the Parkinson's Disease Brain. Int J Mol Sci. 2024;25(11):6168. Published 2024 Jun 3. doi:10.3390/ijms25116168
Comments 8: All over the manuscript: There is a significant lack of hierarchical order of contents. The paragraphs and content have been described with no link and it seems authors have been in a hurry to summarize a large amount of data. It is strongly recommended to speak about the exact types of mutations and their effect both on tumors and NDDs. Also, it is strongly recommended to speak about the statuses of downstream genes which are expressed or inhibited due to each methylation/demethylation. Thus, the manuscript can reach a more reliable and stronger conclusion. The tables are good, figures seem good. Only It is recommended to determine the genes which their methylation is changed during both cancers and NDDs. This determinations can be shown with a special sign or color.
Response 8: We appreciate this reviewer's suggestion and have made major changes to section 5 (HMTs and HDMs in Brain Cancer and Neurodegenerative Diseases). Section 5 was condensed into a description of a few key modifiers (SETD2, G9a/GLP, and KDM2A/KDM2B). We further expanded upon downstream gene targets, when documented evidence was found, and cellular mechanisms in brain tumors and neurodegenerative diseases. To deepen the analysis, we explored the overlap between these modifiers’ mechanisms and pathways involved in neurodegeneration and brain cancer. We removed the paragraphs related to SMYD1, KMT2A, PRC2, PHF8, JMJD2A, and JMJD2C, and moved SUV39H1 to section 4. Additionally, Figure 5’s purpose is to demonstrate the genes that are involved in both brain cancer and neurodegenerative diseases.
Comments 9: Lines 713-715: The statement “However, few modifiers are involved with both brain cancers and neurodegenerative diseases, highlighting the need for more studies” needs more explanations and justifications, since the principal idea of the manuscript was based on the opposite of this statement.
Response 9: Many modifiers have been found to be involved in multiple cancer types. This is demonstrated throughout the manuscript as each modifier is involved across diverse cancer types. Many modifiers also have been found to be involved in brain cancer, and many mechanisms are also found (line 97 and section 3). Furthermore, modifiers are upregulated or downregulated in neurodegenerative disease tissues, but the mechanisms are less known (line 97-99 and section 4). Additionally, histone lysine modifiers are known to be involved in neuron crosstalk and neuron differentiation. These two processes play a role in the development of both brain cancer and neurodegenerative diseases. However, in both brain cancer and neurodegenerative diseases, few modifiers are shown to have a documented overlap, as seen with the number of modifiers in Table 2. Therefore, the principal idea of the manuscript is to review the current research related to the aforementioned concepts and to highlight the need for further studies to advance treatment.
Reviewer 2 Report
Comments and Suggestions for Authors
Title: Epigenetic Modifiers: Exploring the Roles of Histone Methyltransferases and Demethylases in Cancer and Neurodegeneration.
Summary
In this review article, Shilpa S Dhar et al. summarized the role of histone methyltransferases (HMTs) and demethylases (HDMs) in cancers and neurodegenerative diseases, including Alzheimer's, Parkinson's, and Huntington’s diseases. However, this manuscript is not publishable in Biology without revision.
Major comments are listed below.
1. This review article used many abbreviations for the names of histone methyltransferases and demethylases. A list of abbreviations and their full names should be provided.
2. The role of HMTs and HDMs in Brain Cancer and Neurodegenerative Diseases was discussed in section 3, “Roles of HMTs and HDMs in Brain Cancer,” and section 4, “Roles of HMTs and HDMs in Neurodegenerative Diseases.” The authors also present section 5, “Roles of HMTs and HDMs in Brain Cancer and Neurodegenerative Diseases.” What is the purpose of this section? The authors may need to better organize these paragraphs.
3. The authors summarized the role of many HMTs and HDMs in Brain Cancer and Neurodegenerative Diseases and discussed the relevant research for each protein. However, the readers can’t understand their functions. It is better to focus on a few important proteins or genes and provide details on their potential cellular mechanisms in diseases.
4. The authors should provide more of their own insights into the roles of HMTs and HDMs in Brain Cancer and Neurodegenerative Diseases rather than present the results from other published papers. For example, what are the important questions about these enzymes? What are the hot topics in these research areas? Are there any contradictory results in different studies?
Comments on the Quality of English LanguageThe English could be improved to more clearly express the research.
Author Response
Thank you for your valuable comments. We greatly appreciate the insight you have provided to strengthen our manuscript.
Title: Epigenetic Modifiers: Exploring the Roles of Histone Methyltransferases and Demethylases in Cancer and Neurodegeneration.
Summary
In this review article, Shilpa S Dhar et al. summarized the role of histone methyltransferases (HMTs) and demethylases (HDMs) in cancers and neurodegenerative diseases, including Alzheimer's, Parkinson's, and Huntington’s diseases. However, this manuscript is not publishable in Biology without revision.
Major comments are listed below.
Comments 1: 1. This review article used many abbreviations for the names of histone methyltransferases and demethylases. A list of abbreviations and their full names should be provided.
Response 1: A separate document has provided a list of abbreviations.
Comments 2: 2. The role of HMTs and HDMs in Brain Cancer and Neurodegenerative Diseases was discussed in section 3, “Roles of HMTs and HDMs in Brain Cancer,” and section 4, “Roles of HMTs and HDMs in Neurodegenerative Diseases.” The authors also present section 5, “Roles of HMTs and HDMs in Brain Cancer and Neurodegenerative Diseases.” What is the purpose of this section? The authors may need to better organize these paragraphs.
Response 2: The purpose of section 3 is to identify the roles of modifiers involved in brain cancer, but not in neurodegenerative diseases. As stated at the end of section 3, such diseases may use similar pathways, so documented effects of HMTs and HDMs in brain cancer could have implications for neurodegenerative diseases. For section 4, its purpose is identifying modifiers involved in neurodegenerative diseases that have not yet been found to have an effect on brain cancer. The end of section 4 discusses how two processes (neuron crosstalk and neuron differentiation) involved in the development of neurodegenerative diseases also have been implicated to play a role in cancer, demonstrating potential overlap of various pathways. Section 5 discusses modifiers that have documented changes or effects in both brain cancer and neurodegenerative diseases. To further clarify the purpose of these sections, the title of section 5 has changed to “HMTs and HDMs Involved in both Brain Cancer and Neurodegenerative Diseases”.
Comments 3: 3. The authors summarized the role of many HMTs and HDMs in Brain Cancer and Neurodegenerative Diseases and discussed the relevant research for each protein. However, the readers can’t understand their functions. It is better to focus on a few important proteins or genes and provide details on their potential cellular mechanisms in diseases.
Response 3: We appreciate this reviewer's suggestion and have made major changes to section 5 (HMTs and HDMs in Brain Cancer and Neurodegenerative Diseases). Section 5 was condensed into a description of a few key modifiers (SETD2, G9a/GLP, and KDM2A/KDM2B). We further expanded upon downstream gene targets, when documented evidence was found, and cellular mechanisms in brain tumors and neurodegenerative diseases. We removed the paragraphs related to SMYD1, KMT2A, PRC2, PHF8, JMJD2A, and JMJD2C, and moved SUV39H1 to section 4.
Comments 4: 4. The authors should provide more of their own insights into the roles of HMTs and HDMs in Brain Cancer and Neurodegenerative Diseases rather than present the results from other published papers. For example, what are the important questions about these enzymes? What are the hot topics in these research areas? Are there any contradictory results in different studies?
Response 4: To provide our own insights, we explored the overlap between these modifiers’ mechanisms and pathways involved in neurodegeneration and brain cancer. We also discussed potential future studies and their impacts. Regarding contradictory studies, we touched on two studies showing opposite functions of SETMAR in GBM. We also discussed how KDM2A and KDM2B are shown to have different roles in Alzheimer’s disease.
Comments on the Quality of English Language
Comments 5: The English could be improved to more clearly express the research.
Response 5: We have used MD Anderson Scientific Editor's editing services and implemented his suggestions to improve the English and express the research more clearly.
Round 2
Reviewer 1 Report
Comments and Suggestions for Authors
The manuscript has been improved by authors substantially and now it is suitable for publication.
Author Response
Thank you for your review.
Reviewer 2 Report
Comments and Suggestions for Authors
The authors addressed most of my questions.
There are two major issues. First, no figures in the revised version. Second, this review article used many abbreviations for the names of histone methyltransferases and demethylases. A list of abbreviations and their full names should be provided.
Author Response
We have already revised the manuscript with figures and abbreviations list.